# Retention in care and predictors of attrition among HIV-infected patients who started antiretroviral therapy in Kinshasa, DRC, before and after the implementation of the 'treat-all' strategy

Nadine Mayasi[1], Hippolyte Situakibanza[1], Marcel Mbula[1], Murielle Longokolo[1], Nathalie Maes[2], Ben Bepouka[1], Jérôme Odio Ossam[1], Michel Moutschen[3,4], Gilles Darcis[3]*

1 Department of Internal Medicine, Infectious and Tropical Diseases, University Clinics of Kinshasa, Kinshasa, Democratic Republic of the Congo [DRC], 2 Biostatistics and Medico-Economic Information Department, University Hospital of Liège, Liège, Belgium, 3 Department of Internal Medicine and Infectious Diseases, Liège University Hospital, Liège, Belgium, 4 AIDS Reference Laboratory, University of Liège, Liège, Belgium

* gdarcis@chuliege.be

## Abstract

The retention of patients in care is a key pillar of the continuum of HIV care. It has been suggested that the implementation of a "treat-all" strategy may favor attrition (death or lost to follow-up, as opposed to retention), specifically in the subgroup of asymptomatic people living with HIV (PLWH) with high CD4 counts. Attrition in HIV care could mitigate the success of universal antiretroviral therapy (ART) in resource-limited settings. We performed a retrospective study of PLWH at least 15 years old initiating ART in 85 HIV care centers in Kinshasa, Democratic Republic of Congo (DRC), between 2010 and 2019, with the objective of measuring attrition and to define factors associated with it. Sociodemographic and clinical characteristics recorded at ART initiation included sex, age, weight, height, WHO HIV stage, pregnancy, baseline CD4 cell count, start date of ART, and baseline and last ART regimen. Attrition was defined as death or loss to follow-up (LTFU). LTFU was defined as "not presenting to an HIV care center for at least 180 days after the date of a last missed visit, without a notification of death or transfer". Kaplan–Meier curves were used to present attrition data, and mixed effects Cox regression models determined factors associated with attrition. The results compared were before and after the implementation of the "treat-all" strategy. A total of 15,762 PLWH were included in the study. Overall, retention in HIV care was 83% at twelve months and 77% after two years of follow-up. The risk of attrition increased with advanced HIV disease and the size of the HIV care center. Time to ART initiation greater than seven days after diagnosis and Cotrimoxazole prophylaxis was associated with a reduced risk of attrition. The implementation of the "treat-all" strategy modified the clinical characteristics of PLWH toward higher CD4 cell counts and a greater proportion of patients at WHO stages I and II at treatment initiation. Initiation of ART after the

**Data Availability Statement:** All data underlying the findings are provided as part of the submitted article.

**Funding:** The author(s) received no specific funding for this work.

**Competing interests:** No competing interests to declare.

implementation of the 'treat all" strategy was associated with higher attrition (p<0.0001) and higher LTFU (p<0.0001). Attrition has remained high in recent years. The implementation of the "treat-all" strategy was associated with higher attrition and LTFU in our study. Interventions to improve early and ongoing commitment to care are needed, with specific attention to high-risk groups to improve ART coverage and limit HIV transmission.

## Introduction

The proportion of people living with HIV (PLWH) on antiretroviral treatment (ART) significantly increased after the adoption of the "triple 90" goals and the "treat-all" (TA) strategy [1], with 68.4% of PLWH on ART in June 2020 [2]. The TA strategy is based on the recommendation that antiretroviral medicines should be prescribed to people as soon as possible after their HIV diagnosis regardless of their CD4 count. Substantial evidence supports the benefits of early initiation of ART [3–6]. Full realization of these benefits requires patient progression through the cascade of care [HIV testing, diagnosis, link to health services, ART adherence, and viral suppression] with continued commitment to care for life. ART programs must maximize retention and adherence to ART while scaling up to reap the benefits of the TA strategy [7–10]. This is a major challenge in resource-limited settings [11].

The overload of work generated in health care facilities due to increased accessibility of ART, with multiple programmatic challenges, [12] could deteriorate the quality of care and patient follow-up [7, 13]. As ART expands in the age of the TA strategy, gaps in the cascade of care may shift to retention and adherence [14, 15]. Fears have been expressed about the probability of attaining satisfactory ART retention and optimal adherence to therapy following the implementation of the TA strategy, specifically in the subpopulation of asymptomatic PLWH with high CD4 counts. This could mitigate the success of universal ART in resource-limited settings (RLSs) [15–19].

The retention of patients in care is a central pillar of the continuum of HIV care [15, 18, 20, 21]. It involves adherence to ART needed to achieve viral load suppression, which is the basis of individual and societal benefits [21–23]. In contrast, patients who discontinue ART rapidly are at increased risk of ART resistance, morbidity, mortality and transmission [24–27]. Unfortunately, at each step of the cascade of care, substantial proportions of patients are lost to follow-up [28]. Most losses to follow-up (LTFUs) occur in the first year after starting treatment [29]. Data from resource-limited countries suggest that the average retention rate 12 months after treatment initiation ranges from 64 to 94% and can decrease to 60% at 60 months [29]. In a systematic review, ART retention in low- and middle-income countries was 78% at 12 months [30]. The main predictors of LTFU or attrition include advanced HIV disease, male sex, younger age and lower education level [13, 15, 31, 32].

The impact of early ART initiation on the attrition of PLWH in care is more controversial. Several studies have shown a reduction in attrition with rapid initiation of ART [33–35]. Nonetheless, others have reported an elevated risk of LTFU or attrition associated with a shorter time to initiate ART within seven days of HIV diagnosis or the same day [13, 15, 18, 19]. Overall, studies investigating the impact of rapid ART initiation for all infected individuals have thus provided mixed results.

It is crucial for ART programs to assess LTFU or attrition risk factors to improve patient retention in care. Although the subject of retention in care has been analyzed in some other resource-limited countries, it has been poorly studied in the Democratic Republic of Congo

(DRC). In particular, the impact of changed modalities for ART eligibility on maintaining PLWH in care in the DRC has not been examined.

To address these gaps, we documented emerging trends in attrition before and after the implementation of the TA strategy among adolescents and adults in Kinshasa as well as factors associated with it. In other words, the objectives of the study were to better define factors associated with attrition. In particular, we investigated the impact of the TA strategy implementation on the risk of attrition. A better understanding of the causes of attrition would ultimately allow the elaboration of strategies aimed at increasing retention in care, a key element of HIV care.

## Methodology

### Ethics statement

This study was approved by the Scientific Committee and the Research Ethics Committee of the School of Public Health of the Faculty of Medicine of the University of Kinshasa in the DRC (ESP/CE/005/2019). All methods were carried out in accordance with relevant guidelines and regulations. Data were fully anonymized before researchers accessed them and they were analyzed anonymously and in accordance with ethical rules of confidentiality. As the study is retrospective, there was no need for informed consent for patients, which was lifted by the ethics committee.

### Study design

We performed a retrospective cohort study of treatment-naïve PLWH ($\geq$ 15 years old) initiating standard first-line antiretroviral therapy in 85 health facilities in Kinshasa, the capital of the DRC, between January 1, 2010, and December 31, 2019.

### Study setting

The 85 facilities included in the analysis are not the main sources of HIV care in Kinshasa. They represent six health care zones out of 18 that are supported by PEPFAR (President's Emergency Plan for AIDS Relief) and ICAP (International Center for AIDS Care and Treatment Programs). The Global Fund supports other health care zones, for a total of 35 healthcare zones. Nevertheless, the six health care zones represented here cover the four districts of Kinshasa. Moreover, the 85 health facilities included in the study are mostly health care centers offering primary medical care and include HIV care as well. This type of institution is common in Kinshasa. We thus believe that the study population is representative of the general population of HIV infected individuals in Kinshasa.

### Data sources and data collection

Data were accessed from the Electronic ART Database (Tier.Net). TIER.Net is an electronic patient management system with modules to capture patient-level data on HIV counseling and testing and pre-ART and ART services. The use of this tool is recommended in the DRC, and its implementation across the DRC is ongoing. This important tool is already widely used in Kinshasa, thanks to the work of data managers in charge of data collection and data encoding. Mobile data managers play a crucial role to collect paper-based records and to encode in Tier.Net. The implementation of Tier.Net is ongoing in other provinces of DRC.

Sociodemographic and clinical characteristics at ART initiation were collected and recorded in the patient's chart by the health professional and then input into the electronic database designed for this purpose. In principle, deaths are included in Tier.Net. Nevertheless,

the update process is sometimes delayed, and it should be stressed that not all events are captured. LTFU patients are not traced, and it is therefore possible that a proportion of LTFUs is due to unrecorded deaths.

The number of PLWH under follow-up in each center defined the size of the HIV care center, which were categorized into small-, medium- and large-sized centers for <100, 100–500 and > 500 PLWH, respectively.

After treatment initiation, PLWH were under follow-up for the assessment of adherence to ART and ART efficacy and tolerance, according to a schedule established by the National Plan to Fight AIDS, at day 14, 1 month, 3 months, and 6 months and later every six months if the patient was considered stable by the physician. Information on dates of HIV diagnosis, treatment initiation, last visit, and next scheduled clinical visit were programmatic data.

We examined the entire cohort, and we divided the cohort into two cohorts of PLWH depending on the timing of ART initiation:

- P1: from January 1, 2010, to October 31, 2016

- P2: from November 1, 2016, to December 31, 2019,

This corresponds to the periods before (P1) and after (P2) the implementation of the TA strategy in the DRC. Regarding ART eligibility criteria, it should be noted that guidelines evolved over time during P1. Before 2013, ART was initiated for individuals with WHO stage III or IV, individuals with CD4 counts below 350/mm$^3$, and pregnant women. From 2013, ART was initiated in individuals with CD4 counts below 500/mm$^3$.

Guidelines regarding cotrimoxazole prophylaxis and tuberculosis (TB) preventive treatment were also modified. In settings where malaria and/or severe bacterial infections are highly prevalent, cotrimoxazole prophylaxis should now be initiated regardless of CD4 cell count or WHO stage. Regarding TB, adults and adolescents living with HIV who are unlikely to have active TB should receive TB preventive treatment as part of a comprehensive package of HIV care.

During data collection, the most recently documented status for each patient on the date of the last clinic visit was retrieved and recorded as living on ART, deceased or lost to follow-up.

## Definitions of attrition and lost to follow-up

Attrition was defined as death or lost to follow-up. LTFU was defined as "not presenting to the care center for at least 180 days after the date of a last missed visit (clinical visit, refill of drugs, blood test) without a notification of death or transfer" [32]. The date of LTFU was defined as the most recent visit or the day of the initiation of ART if the patient presented only at the initiation visit. Retention was defined as when a participant was alive on ART with no outcome of death or LTFU at the end of data collection. It should be noted that there are no common standardized strategies to reduce LTFUs thus far. Several strategies are implemented to reduce attrition: support from relay persons in the community, peer advisers, and adherence clubs. The implementation of such approaches is recommended by the national plan against HIV/ AIDS. Nevertheless, these strategies are not used in all facilities.

## Statistical analysis

The results are presented as the means and standard deviations (SDs) or medians and interquartile ranges (IQRs) for continuous variables. Qualitative variables were described using frequency tables (numbers and percentages). Kaplan–Meier curves were used to present retention data. Mixed effects Cox regression models were used to examine each prognostic factor for risk of loss to follow-up or death considering the random care center effect. The results

were expressed using hazard ratios (HRs), 95% HR confidence intervals, and p values. In order to meet the proportional hazard assumption, the analysis of the impact of advanced disease and of the timing of ART initiation (P1 or P2) were limited to the first 4 years after treatment initiation. Multivariate models were built including variable with p-value <0.10 in the univariate analysis. Since the analysis of the impact of advanced disease and of the timing of ART initiation (P1 or P2) were limited to the first 4 years after treatment, the multivariate models were also restricted to those first 4 years.

The entire cohort was studied, and then separate models were built for P1 and P2.

The results were considered significant at the 5% critical level (p<0.05). Missing data were not replaced, and calculations were always performed on the maximum number of data available. Data analysis was carried out using SAS (version 9.4 for Windows) and R (version 3.6.1) packages.

## Results

### Patients' characteristics at treatment initiation

Between January 1, 2010, and December 31, 2019, we included 17,318 patients (Fig 1). Patients of unknown age or of age below fifteen and patients previously treated (and thus already linked to care) were excluded from the analysis (Fig 1). Transferred-out patients were kept in the analysis (Fig 1). The final cohort comprised 15,762 PLWH.

Table 1 summarizes the general characteristics of all PLWH at treatment initiation and according to the two periods before (P1) and after (P2) the implementation of the TA strategy. Overall, there were more women (67.4%). The mean age at the initiation of ART was 40 ± 11.4 years. The median CD4 cell count was 247 cells/mm³. A total of 69.8% of patients for which CD4 cell count was available started ART at a late stage (CD4 <350 cells/mm³). 35.3% started ART with advanced HIV disease (AHD), defined as having a CD4 cell count below 200/mm³ and/or WHO stage III or IV HIV. The majority had an initial ART regimen (Table 1 and S1 Fig) based on TDF (81.7%) and EFV (67.1%) and rapidly started ART following diagnosis (83% within 7 days of diagnosis).

The P1 and P2 cohorts included 4481 and 11281 PLWH, respectively. The proportion of women was higher (70.8 vs. 66%, p<0.0001) and PLWH were slightly younger (39.4 vs.

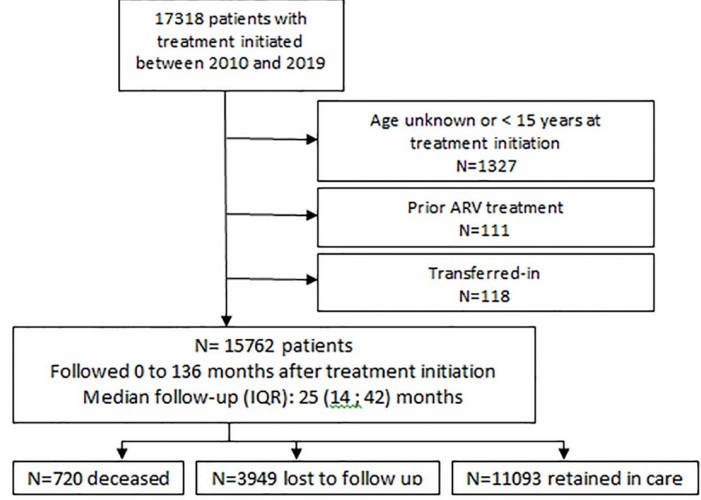

**Fig 1. Flowchart.**

**Table 1. Description of PLWH at treatment initiation [N = 15762].**

| | All | | ARV initiated <Nov 2016 | | ARV initiated ≥Nov 2016 | | Comparison p value |
|---|---|---|---|---|---|---|---|
| | N non missing | Results | N non missing | Results | N non missing | Results | |
| Sex, female | 15762 | 10620 [67.4] | 4481 | 3173 [70.8] | 11281 | 7447 [66.0] | <0.0001 |
| Pregnant women | 10388 | 857 [8.2] | 2983 | 412 [13.8] | 7405 | 445 [6.0] | <0.0001 |
| Age [years] | 15756 | 40.0 ± 11.4 | 4481 | 39.4 ± 11.0 | 11275 | 40.1 ± 11.5 | 0.0004 |
| 15–19 | | 336 [2.1] | | 114 [2.5] | | 222 [2.0] | |
| 20–24 | | 1182 [7.5] | | 302 [6.7] | | 880 [7.8] | |
| 25–49 | | 11129 [70.6] | | 3271 [73.0] | | 7858 [69.7] | |
| ≥50 | | 3115 [19.8] | | 794 [17.7] | | 2321 [20.6] | |
| Height [cm] | 256 | 164 ± 9 | 249 | 164 ± 9 | 7 | 164 ± 9 | - |
| Weight [kg] | 13632 | 60.8 ± 11.2 | 3702 | 61.7 ± 12.2 | 9930 | 60.5 ± 10.8 | <0.0001 |
| BMI [kg/m$^2$] | 245 | 22.4 ± 4.6 | 238 | 22.4 ± 4.5 | 7 | 23.2 ± 6.0 | - |
| CD4 [cells/mm$^3$] | 2432 | 247 [136; 386] | 2257 | 243 [133; 381] | 175 | 296 [187; 483] | - |
| <200 | | 937 [38.5] | | 889 [39.4] | | 48 [27.4] | |
| 200–350 | | 762 [31.3] | | 709 [31.4] | | 53 [30.3] | |
| 351–500 | | 402 [16.5] | | 371 [16.4] | | 31 [17.7] | |
| >500 | | 331 [13.6] | | 288 [12.8] | | 43 [24.6] | |
| WHO stage | 12037 | | 3372 | | 8665 | | <0.0001 |
| I | | 5250 [43.6] | | 850 [25.2] | | 4400 [50.8] | |
| II | | 2749 [22.8] | | 720 [21.3] | | 2029 [23.4] | |
| III | | 3626 [30.1] | | 1621 [48.1] | | 2005 [23.1] | |
| IV | | 412 [3.4] | | 181 [5.4] | | 2311 [2.7] | |
| Advanced HIV[a] | 12492 | 4413 [35.3] | 3770 | 2155 [57.2] | 8722 | 2258 [25.9] | <0.0001 |
| Initial ART | 15738 | | 4465 | | 11273 | | <0.0001 |
| TDF/3TC/DTG | | 2588 [16.4] | | 0 [0.0] | | 2588 [23.0] | |
| TDF/3TC/EFV | | 10031 [63.7] | | 1799 [40.3] | | 8232 [73.0] | |
| TDF/3TC/NVP | | 226 [1.4] | | 125 [2.8] | | 101 [0.9] | |
| TDF/3TC/LPVr | | 34 [0.2] | | 12 [0.3] | | 22 [0.2] | |
| AZT/3TC/DTG | | 1 [0.0] | | 0 [0.0] | | 1 [0.0] | |
| AZT/3TC/EFV | | 529 [3.4] | | 491 [11.0] | | 38 [0.3] | |
| AZT/3TC/NVP | | 2103 [13.4] | | 1986 [44.5] | | 117 [1.0] | |
| AZT/3TC/LPVr | | 22 [0.1] | | 7 [0.2] | | 15 [0.1] | |
| Others | | 204 [1.3] | | 45 [1.0] | | 159 [1.4] | |
| Days since HIV diagnosis | 11732 | 0 [0; 0] | 3017 | 8 [0; 29] | 8715 | 0 [0; 0] | <0.0001 |
| ≤ 7days | | 9738 [83.0] | | 1502 [49.8] | | 8236 [94.5] | |
| > 7days | | 1994 [17.0] | | 1515 [50.2] | | 479 [5.5] | |
| IPT | 14595 | 1955 [13.4] | 3641 | 82 [2.2] | 10954 | 1873 [17.1] | <0.0001 |
| CPT | 15592 | 13459 [86.3] | 4348 | 3328 [76.5] | 11244 | 10131 [90.1] | <0.0001 |

Results are expressed as N [%], mean ± SD, or median [IQR], and p values as chi-square, ANOVA, or Kruskal–Wallis, respectively.

[a] Advanced HIV if CD4<200 cells/mm$^3$ and/or WHO stage III/IV

IPT: isoniazid prophylaxis treatment; CPT: cotrimoxazole prophylaxis treatment

40.1 years, p<0.0001), with a higher proportion of AHD, in the P1 cohort (57.2 vs. 25.9%, p<0.0001). There was a slight decrease in the proportion of adolescents during P2 (2% vs. 2.5%).

As expected, the proportion of PLWH starting ART rapidly was lower during P1 (49.8 vs. 94.5%, p<0.0001) (Table 1 and S2 Fig). The majority of PLWH in the P1 cohort had an initial

TDF/EFV (40.3%) or AZT/NVP (44.5) regimen, and TDF/EFV was dominant in the P2 cohort (73%). A total of 23.3% of the patients in the P2 cohort were treated with a DTG-based regimen (Table 1 and S1 Fig). The ART initiation time was also shorter during P2 (Table 1). Cotrimoxazole (76.5% vs. 90.1%) and INH prophylaxis (2.2% vs. 17.1%) were more frequent during P2 than during P1 (p<0.0001) due to WHO guideline modifications regarding prophylaxis.

### Description of attrition and factors associated with attrition

Patients were under follow-up for a maximum duration of 136 months and a median of 25 months (IQR 14; 42). In total, there were 4,664 cases of attrition, of which 84.6% were due to LTFU.

Fig 2 shows the overall proportion of patients alive and maintaining ART during the study period. The retention (95% CI) reached 89% (88; 89), 83% (83; 84), 77% (77; 78) and 59% (58; 60) at six months, one year, two years and five years of ART, respectively (Fig 2 and S1 Table).

Table 2 shows the analysis of different characteristics associated with attrition (univariate and multivariate models). In the multivariate models, a higher risk of attrition was associated with advanced HIV disease (HR (95% CI) 1.2 (1.1; 1.3), p = 0.0003), ART initiation after November 2016 ("treat all implementation") (HR (95% CI) 1.4 (1.2; 1.5), p<0.0001) and bigger care centers (HR (95% CI) 1.3 (1.1; 1.6), p = 0.0015) (Table 2). In contrast, a lower risk of attrition was associated with a higher weight (HR (95% CI) 0.97 (0.96; 0.97), p<0.0001), greater delay (>7 days) before ART initiation (HR (95% CI) 0.84 (0.74; 0.97), p = 0.013), Cotrimoxazole prophylaxis (HR (95% CI) 0.64 (0.57; 0.71), p<0.0001). In the multivariate model, gender and age did not have a significant impact on attrition.

As expected, the duration of follow-up was longer during P1 (median duration of follow-up 56 months IQR: (29; 71) vs. 22 months IQR: (13; 30)). Overall attrition from 0 to 4 years after treatment initiation was higher in the P2 cohorts (p = 0.0019) (Fig 3 and S2 Table). The risk factors for attrition were broadly similar during the two study periods (S3 Table).

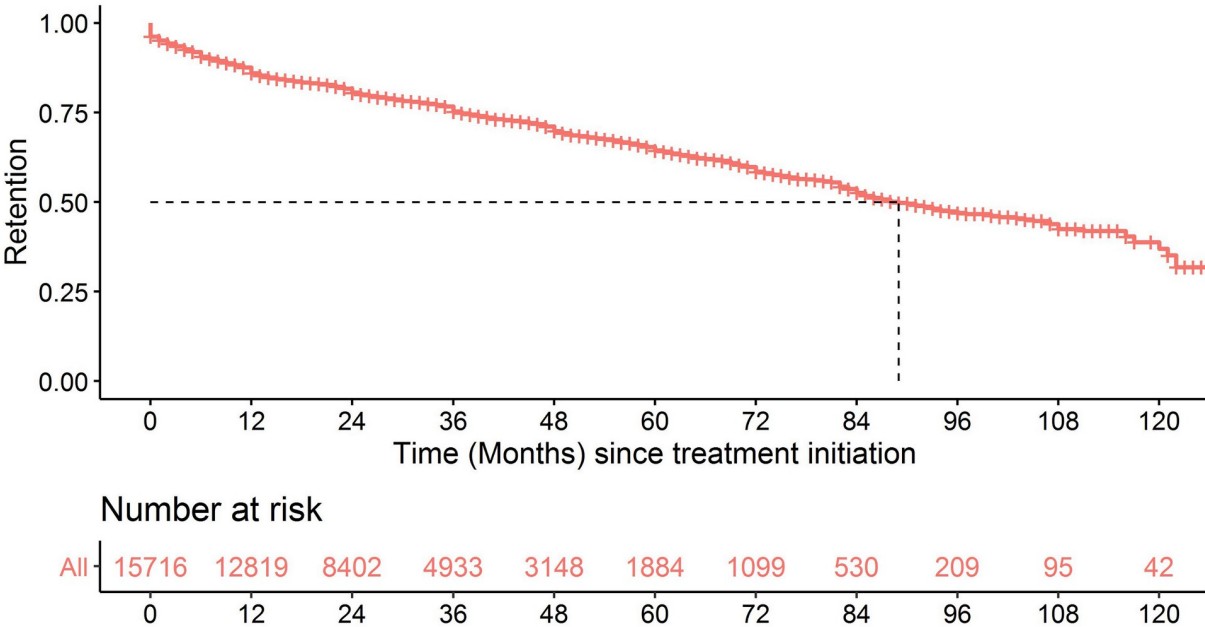

**Fig 2. Kaplan–Meier plot of retention since ART initiation for the entire cohort.**

**Table 2. Characteristics associated with retention: Mixed effects Cox regression models on probability <u>of death or LTFU</u> for the entire cohort.**

| | | Univariate models | | Multivariate model [N = 9922] | |
|---|---|---|---|---|---|
| | N | HR [95%CI] | p-value | HR [95%CI] | p-value |
| Sex, female | 15716 | 0.94 [0.88; 0.998] | 0.044 | 0.95 [0.86; 1.03] | 0.22 |
| Pregnant women | 10351 | 0.98 [0.87; 1.1] | 0.74 | - | - |
| Age <25 years [ref ≥25] | 15716 | 1.1 [1.01; 1.2] | 0.026 | 0.95 [0.82; 1.1] | 0.48 |
| Weigth [kg] | 13586 | 0.97 [0.96; 0.97] | <0.0001 | 0.97 [0.96; 0.97] | <0.0001 |
| CD4 >350 cells/mm$^3$ | 2432 | 0.95 [0.83; 1.1] | 0.51 | - | - |
| WHO stage III /IV [ref = I/II] | 11997 | 1.3 [1.2; 1.3] | <0.0001 | - | - |
| Advanced HIV disease [a] | 12452 | 1.3 [1.2; 1.4] | <0.0001 | 1.2 [1.1; 1.3] | 0.0003 |
| ART initiation before [= ref] or ≥ Nov2016 | 15716 | 1.1 [1.04; 1.2] | 0.0019 | 1.4 [1.2; 1.5] | <0.0001 |
| Days since HIV diagnosis >7 days [ref = ≤7 days] | 11694 | 1.1 [1.02; 1.2] | 0.010 | 0.84 [0.74; 0.97] | 0.013 |
| IPT | 14549 | 0.94 [0.85; 1.04] | 0.21 | - | - |
| CPT | 15546 | 0.66 [0.61; 0.71] | <0.0001 | 0.64 [0.57; 0.71] | <0.0001 |
| Care Center [ref = small] | 15716 | | | | |
| Medium | | 1.3 [1.2; 1.5] | <0.0001 | 1.1 [0.97; 1.3] | 0.13 |
| Big | | 2.1 [1.9; 2.4] | <0.0001 | 1.3 [1.1; 1.6] | 0.0015 |

[a] Advanced HIV if CD4<200 cells/mm$^3$ and/or WHO III/IV

Mixed effects Cox regression model: p-value, HR [95%CI].

IPT: isoniazid prophylaxis treatment; CPT: cotrimoxazole prophylaxis treatment

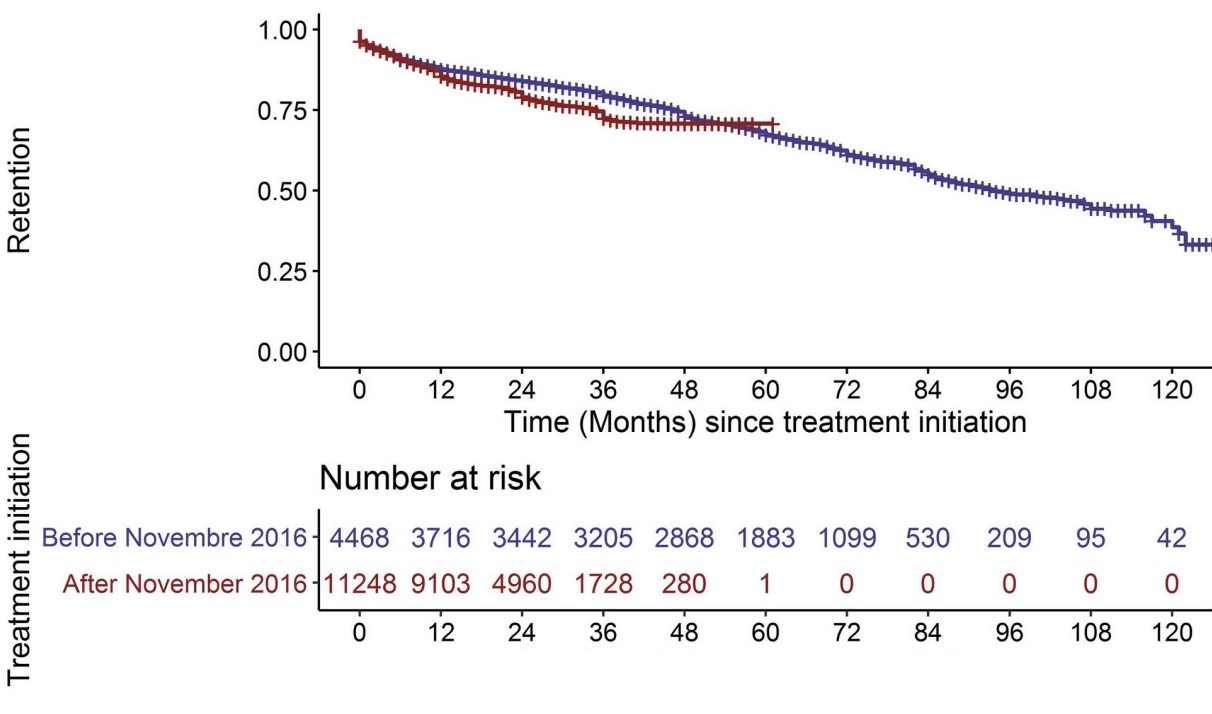

**Fig 3. Kaplan–Meier plots of retention since ART initiation for the P1 and P2 cohorts.** Overall retention from 0 to 4 years after treatment initiation was not significantly different between the P1 and P2 cohorts [Cox regression model: p = 0.0019].

**Table 3. Characteristics associated with retention: Mixed effects Cox regression models on probability of LTFU for the entire cohort.**

| | | Univariate models | | Multivariate model [N = 9922] | |
|---|---|---|---|---|---|
| | N | HR [95%CI] | p-value | HR [95%CI] | p-value |
| Sex, female | 15716 | 0.96 [0.90; 1.02] | 0.22 | - | - |
| Pregnant women | 10351 | 1.0 [0.92; 1.2] | 0.46 | - | - |
| Age <25 years [ref ≥25] | 15716 | 1.2 [1.05; 1.3] | 0.0028 | 0.99 [0.85; 1.2] | 0.94 |
| Weigth [kg] | 13586 | 0.97 [0.97; 0.98] | <0.0001 | 0.97 [0.97; 0.98] | <0.0001 |
| CD4 >350 cells/mm$^3$ | 2432 | 1.1 [0.93; 1.3] | 0.30 | - | - |
| WHO stage III /IV [ref = I/II] | 11997 | 1.1 [1.01; 1.2] | 0.041 | - | - |
| Advanced HIV disease [a] | 12452 | 1.1 [0.99; 1.2] | 0.084 | 0.97 [0.87; 1.1] | 0.55 |
| ART initiation before [= ref] or ≥ Nov2016 | 15716 | 1.3 [1.2; 1.4] | <0.0001 | 1.5 [1.3; 1.7] | <0.0001 |
| Days since HIV diagnosis >7 days [ref = ≤7 days] | 11694 | 1.1 [0.99; 1.2] | 0.070 | 0.87 [0.74; 1.01] | 0.064 |
| IPT | 14549 | 1.0 [0.95; 1.2] | 0.36 | - | - |
| CPT | 15546 | 0.65 [0.60; 0.70] | <0.0001 | 0.60 [0.53; 0.68] | <0.0001 |
| Care Center [ref = small] | 15716 | | | | |
| Medium | | 1.4 [1.2; 1.5] | <0.0001 | 1.2 [0.999; 1.4] | 0.051 |
| Big | | 2.2 [1.9; 2.4] | <0.0001 | 1.5 [1.2; 1.8] | <0.0001 |

[a] Advanced HIV if CD4<200 cells/mm$^3$ and/or WHO III/IV

Mixed effects Cox regression model: p-value, HR [95%CI].

IPT: isoniazid prophylaxis treatment; CPT: cotrimoxazole prophylaxis treatment

### Factors associated with LTFU

The factors associated with death could be different than those associated with LTFU. We therefore investigated factors associated with LTFU after excluding deaths in the statistical analysis (Table 3). In the multivariate model, a higher risk of LTFU was associated with ART initiation after November 2016 ("treat all implementation") (HR (95% CI) 1.5 (1.3; 1.7), p<0.0001) and bigger care centers (HR (95% CI) 1.5 (1.2; 1.8), p<0.0001) (Table 3). In contrast, a lower risk of LTFU was associated with a higher weight (HR (95% CI) 0.97 (0.97; 0.98), p<0.0001) and Cotrimoxazole prophylaxis (HR (95% CI) 0.64 (0.57; 0.71), p<0.0001). In the multivariate model, age, advanced HIV disease and delay before ART initiation (> 7 days) did not have a significant impact on LTFU.

## Discussion

In this report, we analyzed the proportion of attrition before and after the implementation of the TA strategy in Kinshasa as well as factors associated with it. We showed that attrition has remained very high in recent years. Higher attrition was associated with the size of the care center and with ART initiation after the implementation of the TA strategy. We also studied the impact of the TA strategy of LTFU only. The risk of LTFU was associated with ART initiation after November 2016 (TA strategy implementation) as well.

TA strategy is a prodigious achievement in recent years since a greater proportion of PLWH now have access to therapy. However, the TA strategy can be a challenge for ART programs, as more PLWH are under follow-up and on ART, with a risk of health care system overload in RLSs. The implementation of the TA strategy also substantially modified the characteristics of PLWH on ART towards a greater proportion of asymptomatic PLWH with a higher CD4 cell count. It has been suggested that an increased number of patients who initiated ART under the test-and-treat strategy were lost to follow-up [19]. The performance of the ART program before and after recent TA guidelines has not been evaluated in the DRC thus far.

We showed that the overall proportion of patients alive and maintaining ART during the study period was already below 90% six months after treatment initiation and reached 77% two years following ART initiation. Similar percentages have been observed in other countries, such as Kenya or Ethiopia [9, 36]. Regarding factors associated with attrition, our results can be compared with those of other studies [9, 20, 31, 37–39]. We found that advanced HIV disease [WHO stage III/IV] and management in a large HIV care center were associated with higher risk of attrition. Regarding age, it has been previously shown that adolescents are a group at risk for poor outcomes [40–42] and that adolescent-friendly screening interventions are needed to increase adolescents' access to ART programs [43–45]. We observed similar results in the univariate models but the impact of age was lost in the multivariate analysis.

Advanced disease was associated with attrition but not with LTFU, meaning that the impact on attrition is attributable to death. Nevertheless, it should be mentioned that, among PLWH with advanced disease, death may be an underestimated cause of LTFU. Tracing patients lost to follow-up could shed light on their status, but this is extremely difficult to implement in a large cohort in RLSs.

Patients followed in large centers were more prone to attrition and to LTFU, as previously shown by others [15]. Some could speculate that the overload of work for health care providers may reduce the time needed for individualized education, pre-ART preparation and ongoing counseling or support for treatment adhesion. Differentiated patient-centered service delivery (DSD) has been designed for this purpose, with service delivery responding to specific patient needs and health care system challenges encountered in RLSs [46]. Emphasizing the preferences and expectations of PLWH, DSD addresses the contexts and clinical characteristics of clients and aims at individualizing care for patient populations using a public health approach [29, 46]. These models include specific packages of care based on care needs and dependent on the type of service provided, location of service delivery, frequency of service and quality of provider care. DSD would help ART programs improve the accessibility of care to vulnerable populations and key populations [46] and the quality and effectiveness of care [21, 47, 48] in an environment best suited to specific customer needs. The main objective of differentiated care is to rationalize and remove barriers to patient care according to the intensity and level of services required [46]. The differentiation of services for population subgroups, such as pregnant and breastfeeding women, children, adolescents, elderly individuals and key populations, can help improve access to HIV care and care outcomes [49]. Several studies have shown better results [retention and/or viral suppression] in differentiated models [47, 48, 50, 51].

Cotrimoxazole prophylaxis was associated with a reduced risk of attrition and LTFU. One can speculate that providing patients with a global HIV care package and not ART only may increase treatment adherence and trust in the health care system. Another possibility is that patients accepting prophylactic treatment from the beginning are the patients who trust the health care system and are willing to have a sustained follow-up.

The impact of the implementation of the TA strategy on retention is debated. Some authors showed no influence on retention [6], while others reported a positive [20, 35] or a negative impact [11, 18, 52, 53]. Importantly, we demonstrated that the implementation of the TA strategy significantly modified the clinical characteristics of PLWH on ART toward a greater proportion of men, a higher CD4 cell count and a lower clinical WHO stage at treatment initiation. In this regard, our study showed that the risk of attrition slightly increased in PLWH treated very early following diagnosis. Our study suggests that the implementation of the TA strategy as well as early treatment (test and treat) seems to be associated with negative outcomes regarding retention in care. Increased LTFU was associated with the period following the implementation of the TA strategy as well.

An analysis of barriers to continued engagement in care should be considered at both the programmatic and individual levels to improve retention. Intensive supportive interventions need to address retention in care issues for patients initiating ART in the context of the TA strategy. In addition, different patient tracing strategies, such as SMS reminders, phone calls and home visits, would limit attrition [32, 53].

Our study has several limitations. First, we did not consider temporary interruptions in care (on and off ART). We did not track patients lost to follow-up to determine their fate. Our conclusions on retention must thus be cautiously analyzed and interpreted, as some LTFU patients may have been alive and transferred to another center or they may have died. Transfers to other facilities may have been missed if this information was not provided by the patients, as there is no unique identification code shared by all facilities. Additionally, due to the retrospective nature of the study and routine data collection, some variables were missing. In particular, the initial CD4 cell count may have underestimated the proportion of individuals with advanced-stage HIV infection [54]. Finally, our study was performed in the city of Kinshasa. They might not be generalizable in other settings such as less urban area.

## Conclusion

We showed attrition remains a major issue in the recent years. Factors associated with attrition and LTFU included a large HIV care center and ART initiation after the implementation of the TA strategy. Although the benefits of this strategy are huge, the potential negative impact on retention should not be neglected and has to be better studied and understood. Targeted interventions to improve early and ongoing commitment to care are needed to improve ART coverage and limit HIV transmission.

## Supporting information

**S1 Fig. Evolution of ARV treatment at initiation.**
(DOCX)

**S2 Fig. Evolution of the proportion of PLWH who started ART the week of HIV diagnosis.**
(DOCX)

**S1 Table. Retention [N = 15762 PLWH patients].**
(DOCX)

**S2 Table. Retention before and after November 2016 [N = 15762].**
(DOCX)

**S3 Table. Characteristics associated with retention: Mixed effects univariate Cox regression models on probability of death or LTFU for the entire cohort for patients treated before or $\geq$ Nov 2016.**
(DOCX)

## Acknowledgments

We thank all patients and staff at the HIV care facilities included in these analyses.

## Author Contributions

**Conceptualization:** Nadine Mayasi, Hippolyte Situakibanza, Marcel Mbula, Murielle Longokolo, Ben Bepouka, Jérôme Odio Ossam, Gilles Darcis.

**Data curation:** Nadine Mayasi, Hippolyte Situakibanza, Marcel Mbula, Murielle Longokolo, Ben Bepouka, Jérôme Odio Ossam.

**Formal analysis:** Nadine Mayasi, Nathalie Maes, Gilles Darcis.

**Investigation:** Nadine Mayasi.

**Methodology:** Nadine Mayasi, Nathalie Maes, Gilles Darcis.

**Supervision:** Hippolyte Situakibanza, Marcel Mbula, Murielle Longokolo, Ben Bepouka, Jérôme Odio Ossam, Michel Moutschen, Gilles Darcis.

**Validation:** Nathalie Maes, Michel Moutschen, Gilles Darcis.

**Writing – original draft:** Nadine Mayasi.

**Writing – review & editing:** Nathalie Maes, Michel Moutschen, Gilles Darcis.

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
