## [Decision Letter · Decision Letter 0]

11 Aug 2021

 PGPH-D-21-00142 Retention in care and predictors of attrition among HIV-infected patients who started antiretroviral therapy in Kinshasa, DRC, before and after implementation of the ‘Treat-All’ policy. PLOS Global Public Health

Dear Dr. DARCIS,

Thank you for submitting your manuscript to PLOS Global Public Health. After careful consideration, we feel that it has merit but does not fully meet PLOS Global Public Health’s publication criteria as it currently stands. Therefore, we invite you to submit a revised version of the manuscript that addresses the points raised during the review process.

 EDITOR'S COMMENTS: Thank you for your submission to PLOS Global Public Health. Please find below the comments and suggestions of the two reviewers of your paper, with which I agree. In particular the questions regarding the methodology employed (cohort definition, follow-up time, choice of outcomes, adjustment to confounding) are essential items that need to be clarified. ==============================

We look forward to receiving your revised manuscript.

Kind regards,

Sabine Hermans

Academic Editor

Journal Requirements:

Additional Editor Comments (if provided):

Reviewers' comments:

Reviewer's Responses to Questions

**Comments to the Author**

1. Does this manuscript meet PLOS Global Public Health’s publication criteria? Is the manuscript technically sound, and do the data support the conclusions? The manuscript must describe methodologically and ethically rigorous research with conclusions that are appropriately drawn based on the data presented.

Reviewer #1: Yes

Reviewer #2: Yes

2. Has the statistical analysis been performed appropriately and rigorously?

Reviewer #1: I don't know

Reviewer #2: No

3. Have the authors made all data underlying the findings in their manuscript fully available (please refer to the Data Availability Statement at the start of the manuscript PDF file)?

Reviewer #1: Yes

Reviewer #2: No

4. Is the manuscript presented in an intelligible fashion and written in standard English?

Reviewer #1: Yes

Reviewer #2: Yes

5. Review Comments to the Author

Reviewer #1: Review of PGPH-D-21-00142

Summary: Using health records from thousands of patients in Kinshasa, the effect of early vs later ART initiation was explored on retention/attrition of patients.

Major comments:

-I think the premise that the Treat All guidelines may increase healthcare work and perhaps impact ability of clinics to follow-up on patients. Data are mixed and so tthis is the reason for this study. Perhaps this could be outlined more clearly and succinctly in the introduction.

This seems like a valid study to help with this question but there are a number of limitations.

-Is all patient care documented in Tier.Net? is this used across the DRC? Does this also record deaths? If people went to different facilities would it be captured? I think more details on your main data source with be helpful.

-More details on the clinics that these data are taken from would also be helpful—are these the main sources of HIV care in Kinshasa? Where else do people get HIV care from? Are there guidelines about following up clients who miss appointments? Is this standard already?

-Did guidelines change for INH and cotrimoxazole provision? Is there another reason for the large difference between P1 and P2?

-Were you able to determine if attrition was LTFU versus death? Are death’s recorded in a central location?

-I am finding it confusing to understand that some retention was improved but that overall attrition was worse in P2 compared to P1…

-There are some grammatical errors and missing words throughout that would benefit from perhaps a third-party editor.

-why would you think that INH prophylaxis or clotrimaxole would be associated with LTFU?

-there are obviously way more patients in P1 compared to P2—do you think this is a limitation? Should you have tried to match the number of patients in P1 compared to P2? Similarly follow-up time is very different for the populations and could this have impacted your results??

-were any of your analyses adjusted? I could imagine that poor baseline CD4 or functional status could be associated with death or ‘attrition’.

-In results, you refer to figures and tables. I would also write out the results you are trying to highlight from your figures and tables for easy of reading and to really highlight the main results of the paper.

-does it make sense to define attrition as both death and loss to follow-up? These seem like very different measures and have different mechanisms—perhaps the healthier patients are lost to follow-up as they move?

-in figure 4, which line is associated with better retention? The lines cross and then appear to swop positions.

-I think because the sample size is large, you have power to see even very small differences in retention but I think the question is are these clinically relevant and meaningful given such small effect sizes?

Minor comments;

-Please make sure to define abbreviations prior to use: PLWH, DRC.

-Intro: Is the 68.4% of PLWH on ART in June 2020 a global statistic?

-First sentence of second intro paragraph is hard to follow. I think you need to first lay out that there might be work overload with increased ART.

-in methods, what is the difference between the Tier.Net collected data and ‘programmatic data’

-last sentence in discussion is not a complete sentence.

-Probably wouldn’t include coefficient+/- SE in the tables—HRs and 95% CI are more standard.

Reviewer #2: The authors describe ART care outcomes in DRC. It provides programmatic estimates of retention as well as assessment of factors associated with attrition. Findings of this operational research study are important for the context of DRC and this part of Africa and worth publishing.

This article however has several limitations that need improvements. I tried to give some advice below how it could be improved.

Major considerations

While the text is overall written well, it would benefit from language editing and several sections could be more concise.

Abstract: I think better to expand the Result section and shorten the Conclusions. There are a lot of interesting results in the main text that could be presented under results of the abstract.

1 . Introduction:

Define more clearly what treat-all means, such as ART initiation at the time of diagnosis irrespective of immunological criteria.

2. Methods/ study design:

It would be good to have a bit more information about the facilities included in the study. For instance, private vs public sector; supported by other organizations (NGOs, PEPFAR)?

The cohorts of P1 and P2 need better definition. The author mentions that it depended on the timing of ART initiation. I assume this statement refers to treatment eligibility criteria. It is important to know what these treatment eligibility criteria were; for instance CD4<350 vs treat-all (ART initiation irrespective of the CD4 cell count threshold”

3/ Methods/ analysis:

This section should be more clear what happened to patients who were transferred out. It is mentioned later that they have been removed from analysis, but this should already be stated in this section.

Related to TFO, I do not understand why TFO were removed from analysis. Cox regression and Kaplan Meier analysis allows for censoring of TFO and therefore with the benefit of not reducing the sample size.

Throughout the article, it is not clear to me if you calculate crude/unadjusted HRs or present adjusted HR (with all variables included into one model). This needs to be more clear as its interpretation would make a difference.

This part should also describe that one model was build for the entire cohort, and then separate models were build for P1 and P2 (as according to Table 2 and TableS2.

The KM graphs (Figure 3, Advanced HIV disease) suggests that the proportional hazard assumption may not be met. The assumption of the Cox model is that the hazards are proportional (PH), and violation of the PH assumption may lead to biased effect estimates in Cox regression analysis. Did you test for this assumption? If there is a violation, it should be addressed in analysis or should at least be mentioned under limitations.

Overall, the section should be a bit more clear what was really done: Assessment of crude programmatic outcomes (e.g. retention) of the entire cohort and separately for P1 and P2. And evaluation of predictors of all cause attrition (LTFU and death combined) for the entire cohort and separately for P1 and P2. In case you do not want to present all results of these analyses in the main text, it should be more clear what is the main analysis and what is supplementary analysis.

4/ Results:

You report on coef. ± SE and on HR (95% CI). Probably fine not to report on coef. ± SE as the HR is already reported on.

Table 2. What does “Simple cox regression” mean ? Is it univariate or multivariate analysis?

Tables and text: Good to be consistent with reporting of estimates: for instance you report estimates with one, two or three decimal points (see HR) or sometimes with no decimal point. I think better to report the HR with 2 decimal points, proportions with one and to be consistant.

By reporting with three decimal points, I think you want to show that some variables are significantly associated with the outcome. That is not really necessary as you already report on the p values.

Figure 1: Good not to report only the numbers but also the percentages. In addition, the last 2 boxes mention patients deceased and patients LTFU. Add one box for patients retained in care.

Figure 2: re-phrase the title of this Figure. Also mention that it is for the entire cohort.

Figure 3 has 2 titles: Factors associated with attrition vs Retention evolution …

For the Figures with the KM plots, maybe to rename them to something like: Kaplan Meir plots of retention since ART initiation for the entire cohort / for P1 and P2

Table S2: It is not clear why CD4 cell count and WHO staging is not presented in the model/table (see right part of the table for P2).

I think in your analysis it would be more correct to categorize CD4 cell count. For instance, patients with very low CD4 (e.g. deaths) and patients with high CD4 cell count (poor adherence) may be more likely lost to care/ death than patients with a CD4 at around 350. This information may get lost if CD4 is modelled as continues variable, but is important for programme manager to see. The introduction also mentions that patients with high CD4 cell count may be less adherent, so got to have a separate category for them.

Minor comments:

The abbreviation SD for same day ART is only used once, and is the same as used for standard deviations.

This sentence is quite long and could be shortened: “Socio-demographic and clinical characteristics at ART initiation such as sex, age, weight, height, WHO classification, pregnancy, baseline CD4 T cell count, start date of ART, baseline and last ART regimen, Cotrimoxazol and Isoniazid (INH) prophylaxis,” Provide maybe less examples of variables included in analysis as they can also be seen in the tables.

I think the following sentence needs some modification “The date of LTFU was defined as the most recent visit or the day after the start of ART if the patient presented only at the initiation visit.” to “ … or the day of start of ART ….” In case time zero was the day after start of ART, then maybe indicate that it was “…. one day after the start ….”

The Discussion starts with a general discussion about retention and TA. I think it would be better to open the Discussion section with a general statement about what you did in the study and what the main findings were.

Probably better to refer to “CD4 cell count” than “CD4 T cell count”. The latter abbreviation is not commonly used in programmatic papers.

Good to be more consistent in the terms used throughout the paper. For instance: ART vs ARV initiated; treat-all vs Treat All vs Treat-All TA vs Test and Treat.

You need a reference for this sentence: “Patients followed in big centers were more prone to attrition as previously shown by other.”

6. PLOS authors have the option to publish the peer review history of their article (what does this mean?). If published, this will include your full peer review and any attached files.

**Do you want your identity to be public for this peer review?** For information about this choice, including consent withdrawal, please see our Privacy Policy.

Reviewer #1: No

Reviewer #2: No

---

## [Decision Letter · Decision Letter 1]

19 Dec 2021

PGPH-D-21-00142R1

Retention in care and predictors of attrition among HIV-infected patients who started antiretroviral therapy in Kinshasa, DRC, before and after the implementation of the ‘treat-all’ strategy.

Dear Dr. DARCIS,

Thank you for your email. Since your resubmission the manuscript has undergone another round of peer review, which is why it has taken longer than you might have anticipated. Apologies for that. To date, one of the original peer reviewers has not been able to submit his review, so I have done the second review myself.

Thank you for resubmitting your manuscript to PLOS Global Public Health. After careful consideration, we feel that it has merit but does not fully meet PLOS Global Public Health’s publication criteria as it currently stands. Therefore, we invite you to submit a revised version of the manuscript that addresses the points raised during the review process.

EDITOR:

Thank you for submitting your revision. While I can see that the manuscript has improved, I feel there are still some important issues that need addressing before it meets our publication criteria. As there are limited data available on this important issue from your study setting, it would be great to receive an improved version.

Please note that your tracked changes document was not complete (ie not all changes were marked), which has made the review much more time-consuming. For future submissions I would strongly suggest you ensure a complete tracked changes document.

Many of the reviewers’ points were mainly addressed in the rebuttal letter but were hardly implemented in the manuscript. Please could you address these appropriately. These include:

Details about the facilities included in the analysisDetails about tier.net (roll-out, usage, recording of deaths etc)Rationale for not doing multivariable analysisRationale for limitation to 48 months follow-up in P1 vs P2 analysisStudy objectives (last comment of reviewer 3, methods section)First paragraph of discussion still does not contain the summary of the study’s findings, as is common practice in scientific writingThe information in the methods about the study population (its generalizability), data collection is still insufficient. A lot of information was given in the responses, but much of this was not included in the manuscript or only to a limited degree which does not give sufficient insight into risks of selection and misclassification bias.

We look forward to receiving your revised manuscript.

Kind regards,

Sabine Hermans

Academic Editor

Journal Requirements:

Additional Editor Comments (Editor's review comments):

Major comments

Abstract:

The abstract does not include the study objective(s).You phrase your findings in terms of attrition, whereas you present your results in terms of retention. I presume these are the inverse of each other, but this does make it confusing to the reader. I would suggest to stick to the same terminology. This also goes for the rest of the manuscript.You include conclusions about findings you did not present in the results (about adolescents and young adults), please make sure to include

Introduction: please include your study objectives in the last sentences of your last paragraph. They are currently not explicitly stated anywhere. These should cover all of your analyses, and ideally all results will be presented in the order of the objectives.

Methods:

I would suggest to split your study design paragraph into more, it now encompasses much more than just study design (study setting, study population, data collection/data sources, data management/definitions, etc)Please elaborate on the generalizability of the selected HIV care facilities in comparison with the other facilities in Kinshasa. (see also my comment above) It might help to include a study setting paragraph to be able to explain this fully. Please also include the role of ICAP (and PEPFAR) in the treatment programmes – how did the treatment and follow-up differ compared to other HIV care centres? This is important to be able to assess the generalizability of your results.You state that Tier.net implementation is ongoing; could you give an indication of how complete this is? Did your site selection have anything to do with this roll-out? And did all the seleted facilities have complete data entry for all years and for all their patients, or was only part of their population and/or follow-up data entered?Why did you not utilize interrupted time series methodology if your main interest is in comparing P1 to P2?Why did you not utilize multivariable regression analysis? Please include the motivation for this in your methods section. As this likely introduces major bias in your results, this would need to be featured strongly in the limitation section and in the interpretation of your results.Did you adjust for clustering by study site? This would need to be done in all regression analyses.I struggle with the combining death and LTFU as attrition and looking at the risk factors thereof; I would offer that patients who die are no longer at risk of being LTFU, and the factors associated with dying could be different from those associated with LTFU. Unless you have evidence that the majority of LTFU is in actual fact death? Why not look at risk factors for death and LTFU separately?What is the hypothesis behind investigating an association between use of CPT or IPT and retention?

Results:

If your main study objective is to compare P1 to P2 (as the introduction suggests), I would suggest you present your findings stratified by both throughout, and move Figure S2 into the main manuscript. It would then also make sense to present all figures/outcomes curtailing your maximum follow-up period, as defined in your methods.Considering the lack of multivariable analysis, I feel that the results of your univariable analysis are overstated. A lot of these differences could be due to confounding by one or more of the other factors.I cannot find the analysis leading to the “main” P-value of 0.39 in Table S2, which is what the manuscript refers to.

Minor comments

CTX and IPT guidelines: I appreciate that the WHO changed their guidelines, but what was the local guidance on this over time? And what about take up and implementation thereof in practice?

Abstract: Please include how you defined retention, as the inverse of attrition or of LTFU? Please include the method of comparing pre/post.

Methods:

The definition of LTFU is presented in parentheses, please include the source.The term “simple” regression is not standard terminology and can cause confusion (as with reviewer 1), please rephrase to univariable or univariate regression.

Results:

How come you have so many missing CD4 count values, in particular in P2? Did this have anything to do with the data collection and/or change in guidelines? Please make sure to include this information in the methods section, and in the limitation section. (added later after review of reviewer’s comments: please include the citation to your recent paper on this)The line “in settings where malaria… WHO stage” belongs in methods.Couldn’t the information in Tables S1 and S3 not be presented as Kaplan Meier graphs?Was the effect of age linear (Fig 3A suggests otherwise)? If not, I would suggest to include age as a categorical variable. If yes, why did you emphasize the risk of attrition particularly in adolescents in the discussion/conclusion?The inconsistent use of decimal points is confusing to the reader; I agree with the reviewer that it would be better to stick to two for the HR estimates and one for the proportions. By reporting P values you don’t need to show 0.999 instead of 1.0 in order for the reader to understand.

Discussion:

You suggest that taking CPT increases attrition possibly due to patients who accept prophylactic treatment “from the beginning” are more likely to be in follow-up. This suggests that this effect would be stronger in P1 when patients were started on CPT before ART? However, the analysis stratified by P1 and P2 shows no major difference in those estimates. Maybe you could consider editing? (As above, I would suggest many of these effects are due to confounding so I am hesitant to overinterpret them anyway.)

Reviewers' comments:

Reviewer's Responses to Questions

**Comments to the Author**

1. If the authors have adequately addressed your comments raised in a previous round of review and you feel that this manuscript is now acceptable for publication, you may indicate that here to bypass the “Comments to the Author” section, enter your conflict of interest statement in the “Confidential to Editor” section, and submit your "Accept" recommendation.

Reviewer #1: (No Response)

2. Does this manuscript meet PLOS Global Public Health’s publication criteria? Is the manuscript technically sound, and do the data support the conclusions? The manuscript must describe methodologically and ethically rigorous research with conclusions that are appropriately drawn based on the data presented.

Reviewer #1: Partly

3. Has the statistical analysis been performed appropriately and rigorously?

Reviewer #1: I don't know

4. Have the authors made all data underlying the findings in their manuscript fully available (please refer to the Data Availability Statement at the start of the manuscript PDF file)?

Reviewer #1: Yes

5. Is the manuscript presented in an intelligible fashion and written in standard English?

Reviewer #1: Yes

6. Review Comments to the Author

Reviewer #1: Some of my previous comments have been addressed, but often more in the letter than in the actual manuscript.

1. The large # of patients is a strength. The authors state that there are limited other data about HIV retention in the DRC and so it is adding to the literature.

2. My previous comment about what programmatic data and tier.net data are, was not adequately addressed in this resubmission.

3. Why did you look at cotrimoxazole prescribing and its risk of attrition? What was the hypothesized mechanism a priori? If you look for enough associations, you will likely discover some by chance.

4. One of your points in the discussion was that more adolescent friendly services are needed to support ART in this population but I do not find specifically worse outcomes in adolescents in the main results. I think this is coming from a figure, but you should spell this out in the text of the results.

5.Would also reframe the first sentence of the discussion to be specifically what you found, then comparing what you found to what the literature says, strengths, limitations, conclusion. I get a bit lost in what the main results of the paper are given that so much is discussed in both the results section and in the discussion.

6.In the discussion, the term ‘modified’ was used—this term is typically being used in the context of effect modification when results are stratified but I don’t think this is what you meant.

7. It gets a little hard to follow when moving back and forth between attrition/LTFU, when your manuscript is defining attrition as both LTFU and death. Also then figures are talking about retention but also attrition. I would pick either pick retention or attrition/LFTU or death also focusing on one direction—higher risk, lower risk of attrition.

8.“Information on dates of HIV diagnosis, treatment initiation, last visit, and next scheduled clinical visit were programmatic data.” Does that mean it was in Tier.net or captured elsewhere?

7. PLOS authors have the option to publish the peer review history of their article (what does this mean?). If published, this will include your full peer review and any attached files.

**Do you want your identity to be public for this peer review?** For information about this choice, including consent withdrawal, please see our Privacy Policy.

Reviewer #1: No

---

## [Editor Report · Decision Letter 2]

13 Feb 2022

Retention in care and predictors of attrition among HIV-infected patients who started antiretroviral therapy in Kinshasa, DRC, before and after the implementation of the ‘treat-all’ strategy.

PGPH-D-21-00142R2

Dear Dr. DARCIS,

Thank you for your thorough revision of your manuscript 'Retention in care and predictors of attrition among HIV-infected patients who started antiretroviral therapy in Kinshasa, DRC, before and after the implementation of the ‘treat-all’ strategy.', which indeed is now much improved. We are pleased to inform you that it has been provisionally accepted for publication in PLOS Global Public Health.

Best regards,

Sabine Hermans

Academic Editor